# The Effect of the Stirring Speed on the In Vitro Dry Matter Degradability of Feeds

Sonia Tassone [1], Chiara Sarnataro [2,*], Sara Glorio Patrucco [1], Sabah Mabrouki [1] and Salvatore Barbera [1]

1  Department of Agriculture, Forestry, and Food Sciences, University of Turin, 10095 Grugliasco, Italy
2  Department of Agricultural, Food, Environmental and Animal Sciences, University of Udine, 33100 Udine, Italy
*  Correspondence: chiara.sarnataro@uniud.it; Tel.: +39-0432-558589

**Abstract:** In vitro methods have been standardized and tested to correctly simulate the rumen environment and fermentation process. A few studies have verified that the feed degradability achieved as a result of stirring the samples is higher when the samples are incubated under continuous stirring than when they are only stirred twice daily. The objective of this study has been to verify the effect of the speed of stirring on feed degradability during In vitro incubation. For this purpose, the apparent and true dry matter degradability (ADMD and TDMD) of grass hay, pelleted alfalfa, corn silage, barley meal, straw, and a total mixed ration (TMR) were measured after 48 h of incubation in jars under different rotation speeds. The same types of feed were placed in the four jars of each instrument, and the rotation system of the machine was modified to ensure the simultaneous rotation of a pair of original jars (which sometimes stopped and/or rotated slowly and irregularly) together with a pair of modified jars under regular and continuous rotation. A rev counter data logger was mounted onto the jars, and the rotation speeds of the original and modified jars were measured and compared under different conditions (empty jars, jars with liquid, jars with rumen fluid, and sample bags). The modifications to the instruments stabilized the rotation of the jars, thereby making the stirring more regular during incubation. The degradability was partly influenced by the regular stirring, albeit with just one instrument, and for grass hay, barley meal, corn silage, and TMR. In short, it has been found that the regular stirring of sample bags is not essential to obtain reliable degradability measurement during incubation, although it is better to maintain a constant rotation to ensure a regular and standardized In vitro incubation process and therefore to allow reproducibility and comparisons of the results on feed degradability.

**Keywords:** in vitro incubation; stirring speed; feed degradability; Ankom Daisy[II] Incubator

## 1. Introduction

The rumen is the most voluminous digestive organ in ruminant animals, and the microbial fermentation that occurs there is largely responsible for the fermentation and degradability processes that take place there [1].

A great deal of research has focused on using in vivo or in situ techniques which require large amounts of feed, a considerable number of animals, and are highly time-consuming [2]. Conversely, In vitro methods represent an ideal laboratory, and they can be used more precisely and rapidly as alternatives to elucidate the fundamental principles of microbial ecology and to test the effects of the microbiota on the degradation of feeds.

Different In vitro techniques [3,4] and methods [5–7] have been developed to simulate the rumen environment and fermentation.

However, it is only with the development of the Ankom Daisy[II] incubator (AD[II]; Ankom Technology Corporation, Fairport, NY, USA) that it has been possible to automate and standardize the incubation process [8]. The AD[II] simulates in vivo digestion within a

thermostatically controlled chamber, which contains four rotating jars that can simultaneously hold up to 92 samples closed in single bags, together with the inoculum (rumen fluid or feces) and the buffer solution.

Many trials have been conducted to verify the reliability of the instrument as well as to define and standardize the method. For example, filter bags have been found advantageous, as they avoid the need of filtrating the sample [9]; the porosity of filter bags and their interaction with the grinding size of the sample has been defined [10–14]. Moreover, various authors have considered the size of the sample [15–20]. Tests have been conducted to verify the best type of inoculum, for example on rumen fluid and fresh feces [21–25]; on donor species [12,26–29]; on the collection system (fistulated, cannulated, or slaughtered animals) [30–33]; on the time and temperature of storage [17,32–36]; and on the inoculum preparation [37]. Indications have been given on the incubation time [9,38,39]. A few authors have also investigated digestibility within and between jars and considered different buffer solutions; the influence of the position of the jars in the fermenter and the effect of continuous shaking of samples have also been investigated [40,41].

However, no study has yet considered the effect of the speed of rotation of the jars, which often undergo discontinuous stirring, as they sometimes slow down or even stop rotating during the incubation process [42].

Therefore, the objective of this study has been to verify the effects of sample stirring and the rotation speed on the degradability of feeds during a 48 h incubation period.

## 2. Materials and Methods

The study was conducted on two different $AD^{II}$ incubators of the same model type (Ankom Technology Corporation, Fairport, New York, NY, USA), which are normally used in two different laboratories (AD1 and AD2), where they have been in use for several years, to study feed degradability [8,43–45]. The two machines were placed in the same laboratory for the duration of the trial to ensure the same boundary conditions and to use the same inoculum. Each $AD^{II}$ session lasted 48 h, which is the time normally used for an incubation session.

The $AD^{II}$ incubator consists of a thermostatically sealed chamber into which four rotating digestion jars are placed, which are numbered, starting from the first one on the top left and moving clockwise, as positions 1, 2, 3, and 4 (Figure A1).

A kit was specifically developed to ensure efficient and continuous rotation of the two instruments. The kit allows all the jars, or pairs of jars (upper and lower), to rotate continuously, according to the experimental needs. A rev counter data logger was specifically designed and built to count the rounds of each jar. A magnet was positioned on the side of the lid of each jar, and 4 Hall effect sensors were mounted onto the $ADs^{II}$ to detect the magnetic field (Figure A1). Arduino® software was implemented to record the time of each passage of each magnet on an SD memory card.

### 2.1. Degradability of Feeds

Five common ruminant nutrition feeds (grass hay, pelleted alfalfa, corn silage, barley meal, and straw) and a total mixed ration (TMR) were dried, milled (0.5 mm ), and weighed (0.5 g) in quadruplicate in Ankom F57 bags, and incubated in jars 1 and 3, which contained rumen fluid (RF) that had been collected from slaughtered bulls [43] and mixed with a buffer solution [46] for 48 h at 39.5 °C under anaerobic conditions. The RF was collected at a slaughterhouse from four culled dairy cows coming from the same farm and fed with a TMR based on corn silage. The buffer solution was obtained by mixing two solutions in a 5:1 ratio (solution A: $KH_2PO_4$ 10 g/L, $MgSO_4 \cdot 7H_2O$ 0.5 g/L, NaCl 0.5 g/L, $CaCl_2 \cdot 2H_2O$ 0.1 g/L, $CH_4N_2O$ 0.5 g/L; solution B: $Na_2CO_3$ 15.0 g/L, $Na_2S \cdot 9H_2O$ 1.0 g/L). The inoculum was prepared by mixing 400 mL of rumen fluid with 1600 mL of buffer solution. The mixture was purged with $CO_2$. Jars 2 and 4 were filled with the same rumen fluid quantity to allow the correct movement of the instrument. All the jars were removed at the end of incubation, and the fluid was drained off. The bags were rinsed and weighed after drying at 102 °C for

4 h. The bags were later analyzed using an Ankom[200] Fiber Analyzer (Ankom Technology Corporation, Fairport, USA) for neutral detergent fiber (NDF) [47].

The incubation was repeated twice for each jar (1 and 2; 3 and 4), with the same combination of conditions: the modified rotation system up and original rotation system down, and vice versa. The rotation speed of the jars was measured during the incubation of the samples.

### 2.2. Rotation Speed of the Jars

The instrument should have a constant rotation speed of about 60 rounds per minute during operation when the jars contain the bags and are filled with liquid. The rotation speed was checked under two different conditions to check the efficiency of the instrument and identify the causes of slowdowns and/or stops: with empty jars and then with jars filled with 2 L of water. This was done because the weight of the filled jars was hypothesized to be the factor that caused the slowdowns. The rotation speeds of each jar (in the two conditions) were measured on the instruments in the original rotation mechanism (Original) provided by the manufacturer, and the measurement was repeated three times. The same protocol was used after modifying the original rotation mechanism (Modified) to make it less sensitive to changes in the weight of the jars.

The rotation speed of the jars was also measured during sample incubation, with the jars filled with 2 L of RF and the sample bags. In this case, either the lower or the upper part was modified so that the original and modified rotation mechanisms were simultaneous in each of the two instruments. This measure was introduced to control the effect of the variation in the rumen fluid, and it was repeated twice.

### 2.3. Measured Parameters

The degradability was expressed as the In vitro apparent dry matter degradability (ADMD) and true dry matter degradability (TDMD), and was calculated as follows:

$$\text{ADMD } (\%DM) = \frac{DM_{0h} - DM_{residue}}{DM_{0h}} * 100$$

$$\text{TDMD } (\%DM) = \frac{DM_{0h} - DM_{after\ ND\ treatment}}{DM_{0h}} * 100$$

where:

$DM_{0h}$ (%) = dry matter, ante incubation

$DM_{residue}$ (%) = dry matter, post incubation

$DM_{after\ ND\ treatment}$ (% DM) = dry matter after the neutral detergent treatment

The rotation speed of the jars was expressed as the number of average rounds per hour (*rph*), the slowest round per hour, and as the delay (%), which was calculated as:

$$Delay_i\ (\%) = \frac{rph\ empty\ jar_i - rph\ full\ jar_i}{rph\ empty\ jar_i} * 100$$

where *i* was the number of the *jar*, i.e., 1 to 4.

Delay was used to overcome the small manufacturing differences in the diameter of the rollers and pulleys of the AD[II] instruments, which caused differences in the rotation speed.

### 2.4. Statistical Analysis

The ADMD and TDMD, the number of rounds per hour (rph), and the percentage of delay with respect to the empty jars were processed, and the results were discussed. The obtained data were analyzed with SAS [48], and a GLM procedure was used to evaluate the effects of the modification (original vs. modified) and of the instruments (AD1 and AD2) on the degradability of each feed.

## 3. Results

### 3.1. Degradability of Feeds

Table 1 shows a comparison of the degradability of the different feeds and the TMR obtained for the two instruments (AD1, AD2) in their original and modified versions. Moreover, the rotation speed and the delay of the original and modified versions of each instrument were also compared.

**Table 1.** Comparison of the two instruments (AD1, AD2) for the original and modified rotation systems after 48 hours of incubation of the apparent and true dry matter degradability (N = 64), the rotation speed, and the delay (N = 23,690 rph detected).

| Feed | AD1 | | AD2 | | MSE |
|---|---|---|---|---|---|
| | **Original** | **Modified** | **Original** | **Modified** | |
| | Apparent dry matter degradability (ADMD, % DM) | | | | |
| Grass hay | 51.2[A] | 53.0[A] | 35.4[C] | 41.2[B] | 18.23 |
| Barley meal | 72.1[aA] | 77.8[aA] | 59.6[B] | 71.4[bA] | 38.62 |
| Pelleted alfalfa | 49.5[A] | 50.7[A] | 40.4[B] | 44.0[B] | 19.02 |
| Straw | 32.6[A] | 35.9[A] | 13.7[B] | 16.7[B] | 38.52 |
| Corn silage | 56.2[aA] | 60.1[A] | 50.5[bB] | 57.3[A] | 34.42 |
| TMR | 57.6[A] | 60.5[aA] | 47.4[B] | 56.1[bA] | 19.31 |
| | True dry matter degradability (TDMD, % DM) | | | | |
| Grass hay | 64.5[A] | 66.6[A] | 50.0[C] | 55.1[B] | 17.37 |
| Barley meal | 85.4[aAB] | 86.9[A] | 82.6[bB] | 85.6[aAB] | 6.79 |
| Pelleted alfalfa | 63.6[A] | 64.9[A] | 56.7[bB] | 59.5[aB] | 7.73 |
| Straw | 41.8[A] | 44.4[A] | 25.9[B] | 28.8[B] | 30.16 |
| Corn silage | 66.3[aAB] | 69.1[A] | 61.5[bB] | 67.6[A] | 20.09 |
| TMR | 68.3[abA] | 70.6[aA] | 59.9[B] | 66.8[bbA] | 16.66 |
| Rotation speed (rph) | 38.4[D] | 59.4[B] | 47.7[C] | 66.9[A] | 32.91 |
| Delay (%) | 32.2[A] | 0.3[C] | 5.8[B] | −0.5[D] | 119.44 |

[a, b] *p* = 0.05, [A, B, C, D] *p* =< 0.01: based on Tukey's test in the same row. Abbreviations: AD1, Ankom Daisy[II] 1; AD2, Ankom Daisy[II] 2.

When considering the ADMD, the average values of the grass hay, barley meal, pelleted alfalfa, straw, corn silage, and TMR were 45.2 %, 70.2 %, 46.2 %, 24.7 %, 56.0 %, and 55.4 %, respectively.

The degradability measured with AD1 was generally higher than that of AD2, particularly for some feeds, such as, for example, straw (ADMD: 32.6 vs. 13.7 and 35.9 vs. 16.7 %, respectively, for the original AD1 vs. AD2 and modified AD1 vs. AD2).

The ADMD values were always lower when the feeds were digested with the original system, and these differences were significant, albeit just in AD2. A higher ADMD was found in the AD2 modified version for grass hay, barley meal, corn silage, and TMR, although similar values were obtained for AD1, except for grass hay. The pelleted alfalfa and straw were not influenced by the rotation speed of either system.

The average TDMD values were 59.0 % for grass hay, 85.1 % for barley meal, 61.2 % for pelleted alfalfa, 35.2 % for straw, 66.1 % for corn silage, and 66.4 % for TMR.

A modification of the rotation system increased the TDMD in several cases. However, like the ADMD in AD1, the differences between the systems were not significant. The degradability measured with AD2 showed significantly higher values for the modified system, with *p* < 0.01 for grass hay, corn silage, and TMR, and with *p* < 0.05 for pelleted alfalfa. The original AD1 and AD2 systems showed a similar TDMD for barley meal and corn silage.

### 3.2. Rotation Speed of the Jars

When considering the average value of the rotation speed of the four jars, it was found that the modified systems of both instruments had higher speeds (Table 1). The rotation speed decreased significantly in this order: AD2 modified system, AD1 modified system, AD2 original system, and AD1 original system. The average delay of the two instruments was very high for the original rotation mechanism (Table 1). The jars in the original AD1 showed the longest delay (32.2%), while the original AD2 showed 5.8 %. The delay was almost zero for the modified instruments.

The results of the measurement of the rotation speeds (mean and standard deviation, mean $\pm$ SD) of the jars of the two instruments for the original and modified rotation systems and under different conditions (empty jars, with liquid, with RF, and samples) are reported in Table 2. Moreover, the slowest round (rph) and delay (%) of the jars of the original and modified AD1 and AD2 instruments are also reported.

**Table 2.** Average rotation speed, standard deviation (SD), slowest round /h, and delay mean (%, compared with the empty jars) measured for the jars in the two AD[II] incubators over 48 hours with different contents (empty; 2 L of water or rumen fluid (RF); 2 L of RF mixed with a buffer solution plus sample bags; N = 304,736 detected rph).

| Parameter | Content | System | AD1 | | | | AD2 | | | |
|---|---|---|---|---|---|---|---|---|---|---|
| | | | Jar1 | Jar2 | Jar3 | Jar4 | Jar1 | Jar2 | Jar3 | Jar4 |
| Speed mean (rph) | Empty | Original | 56.4 | 56.3 | 57.1 | 56.9 | 50.2 | 49.9 | 53.4 | 53.3 |
| | | Modified | 58.3 | 58.4 | 61.2 | 61.0 | 64.3 | 63.7 | 67.2 | 67.2 |
| | Liquid | Original | 30.4 | 34.3 | 47.4 | 43.2 | 26.8 | 46.6 | 49.6 | 49.3 |
| | | Modified | 57.1 | 57.9 | 59.8 | 59.5 | 65.6 | 66.5 | 67.4 | 66.8 |
| | RF + bags | Original | 45.0 | | 31.6 | | 48.7 | | 42.0 | |
| | | Modified | 59.1 | | 59.7 | | 66.8 | | 66.9 | |
| Speed SD (rph) | Empty | Original | 0.77 | 0.78 | 1.63 | 1.67 | 0.55 | 0.47 | 0.47 | 0.47 |
| | | Modified | 1.65 | 1.68 | 0.63 | 0.60 | 0.56 | 0.63 | 0.63 | 0.62 |
| | Liquid | Original | 7.69 | 9.65 | 3.02 | 8.34 | 16.04 | 7.89 | 0.53 | 4.16 |
| | | Modified | 1.56 | 1.68 | 0.79 | 1.21 | 0.42 | 0.94 | 0.62 | 1.05 |
| | RF + bags | Original | 5.96 | | 6.47 | | 16.44 | | 18.94 | |
| | | Modified | 0.85 | | 2.12 | | 2.30 | | 1.11 | |
| Slowest round (rph) | Empty | Original | 27.9 | 27.9 | 50.7 | 50.7 | 42.9 | 41.9 | 44.4 | 44.4 |
| | | Modified | 55.4 | 55.4 | 58.1 | 57.1 | 63.2 | 63.2 | 65.5 | 66.7 |
| | Liquid | Original | 13.6 | 0.2 | 40.9 | 1.1 | 0.1 | 0.1 | 46.8 | 0.1 |
| | | Modified | 52.9 | 26.3 | 57.1 | 0.1 | 64.3 | 10.7 | 65.5 | 4.8 |
| | RF + bags | Original | 0.2 | | 0.0 | | 0.1 | | 0.0 | |
| | | Modified | 26.5 | | 0.1 | | 0.0 | | 4.8 | |
| Delay mean (%) | Liquid | Original | 46.1 | 39.1 | 17.0 | 24.0 | 46.5 | 6.7 | 7.2 | 7.5 |
| | | Modified | 2.1 | 0.7 | 2.4 | 2.6 | −2.0 | −4.5 | −0.3 | 0.5 |
| | RF + bags | Original | 20.2 | | 44.7 | | 2.9 | | 21.4 | |
| | | Modified | −1.2 | | 2.4 | | −3.9 | | 0.5 | |

Abbreviations: rph, round per hour; RF, rumen fluid; AD1, Ankom Daisy[II] 1; AD2, Ankom Daisy[II] 2.

#### 3.2.1. Original Rotation System

Empty jars—In the original ADs[II], the speed of the empty jars was similar in AD1 and AD2, that is, on average 56.6 $\pm$ 1.28 rph and 51.8 $\pm$ 1.73 rph, respectively. The absolute difference in speed between the two instruments was due to the small differences in the diameters of the free and drive rollers. Slowdowns were observed for the lowest rph, that is, of around 28 $\pm$ 42 rph in AD1 and in AD2, respectively, but the low standard deviations indicated that there were only occasional slowdown episodes.

Jars filled with liquid—The number of rounds for the jars filled with liquid decreased for both instruments. The speed was similar in the pairs and always higher in the lower jars (3 and 4). Jar 1 in AD2 was very slow, and its speed was more variable than that of jar 2, with values of $26.8 \pm 16.04$ and $46.6 \pm 7.89$ rph, respectively. The slowest round in the jars filled with liquid was 0.1 rph, although there were large differences for the same pair, thus indicating jar sliding had taken place on the drive roller. When considering the delay of the jars filled with 2 L of liquid, the average was $42.1 \pm 16.12\%$ in the upper jars (1 and 2) in AD1 and $21.1 \pm 12.20\%$ in the lower ones (3 and 4). The delay in AD2 was very short for jars 2, 3, and 4 (average 7.1%), but very long for jar 1 (46.5%).

Jars filled with rumen fluid and the bags—The number of rounds in jars filled with rumen fluid and bags decreased in both instruments compared with the empty jars, and both instruments recorded stops of 0 rph. The delay for the jars was variable and erratic in the two instruments.

### 3.2.2. Modified Rotation System

*Empty jars*—The speeds of the empty jars in the modified ADs[II] (Table 2) were similar for the original and modified couple. The absolute difference in speed was due to the small differences in the diameters of the free and drive rollers, and the low standard deviations indicated a regular rotation.

Jars filled with liquid—The rph of both instruments remained constant for the jars filled with liquid, and only occasional slowdowns occurred. The speed was similar for the pairs and always higher for the lower jars (3 and 4).

Jars filled with rumen fluid and bags—Only one jar was used per pair (1 and 3). The speed of rotation of the jars filled with RF and the bags also remained constant in both instruments, and only occasional slowdowns were observed. The speed was constant, although a slight and variable delay was observed.

The delays of the original and modified AD[II] incubators are reported in Table 3. All the comparisons were significantly different, due to the large number of recorded laps. The original system had delays ranging between 2.9 and 46.5 %, as a result of the great variability in the rotation. The delays for the modified system for both the liquid and the RF + bags were instead between $-4.5$ and 2.4 %.

**Table 3.** Comparison of the LSMeans delay (%) of the jars, according to the type of content, for the original and modified ADs[II] after 48 hours of rotation (Liquid: DFE= 152045, MSE 55.29; rumen fluid (RF) +bags: DFE= 23683, MSE 87.75).

| Content | System | AD1 | | | | AD2 | | | |
|---|---|---|---|---|---|---|---|---|---|
| | | Jar 1 | Jar 2 | Jar 3 | Jar 4 | Jar 1 | Jar 2 | Jar 3 | Jar 4 |
| Liquid | Original | 46.1[A] | 39.1[B] | 17.0[D] | 24.0[C] | 46.5[A] | 6.7[C] | 7.2[B] | 7.5[B] |
| | Modified | 2.1[B] | 0.7[C] | 2.4[AB] | 2.6[A] | $-2.0$[C] | $-4.5$[D] | $-0.3$[B] | 0.5[A] |
| RF + bags | Original | 20.2[B] | | 44.7[A] | | 2.9[B] | | 21.4[A] | |
| | Modified | $-1.2$[B] | | 2.4[A] | | $-3.9$[B] | | 0.5[A] | |

[A, B, C, D] *p* =< 0.01: based on Tukey's test by Content, System, and Instrument.

## 4. Discussion

In vitro degradability measurement procedures have improved over time and become standardized, thereby ensuring the repeatability and reliability of the results [8]. As far as the stirring of feed samples during incubation is concerned, it is known that if continuous movement is ensured, dry matter digestibility increases [40]. However, many authors have verified that regular and continuous movement of the samples is not always guaranteed [42]. By modifying the rotation system of a pair of jars in AD incubators in an attempt to establish a regular rotation speed, it was verified that the rotation speed had an effect on dry matter degradability. In general, the values of degradability measured for four feeds and of a TMR analyzed with the original AD1 jars and the modified jars of the two instruments were

similar to the values found by other authors [27,49,50], except for straw. Martinéz et al. [51] studied the effect of soaking grass hay on digestibility using a gas production technique, and they found that the dry matter decreased as the soaking time increased. They found values that ranged between a minimum of 21 % and a maximum of 58 %. The ADMD they measured with an AD instrument was 51 % [27] and the TDMD was 69.6 % [20]. Digestibility measured on barley meals has been found to decrease as the particle size increases (from 0.11 to 2.98 mm), that is, from 97.4 % to 72.1 after 7 h of incubation [27]. The digestibility of the first and second cuts of pelleted alfalfa was 58.8 % [50]. The authors showed that pelleting the first cut of alfalfa not only reduced the production of methane but also dry matter digestibility whenever any influence was exerted on pelleting the second cut. The In vitro dry matter digestibility of corn silage, when different sources of lactic acid bacteria were used, ranged between 545 and 645 g/kg [52]. Holden [27], using an AD instrument, measured an average degradability of 63 %. The TDMD of corn silage measured by Cattani et al. [20] was higher ($73 \pm 80.6\%$) than ours ($61.5 \pm 69.1\%$). The TDMD of the TMR measured with the AD instrument was 69% on average [27]. Our ADMD values for straw were much lower than those found in other studies. Yalçin et al. [53] measured the apparent dry matter degradability of wheat, barley, oats, and rice straw in situ and found values that ranged from 440 to 560 g/kg in sheep. The TDMD values obtained for AD1 were similar to those of wheat straw (46.6%) [20].

The degradability of feeds was partially influenced by the rotation speed of the jars. The positive effect of the constant rotation speed, which was guaranteed by the proposed changes in the rotation system, was only observed for one instrument (AD2) and resulted in an increase in the degradability of all the analyzed feeds and TMR, except for the ADMD of the alfalfa and straw pellets and the TDMD of the pelleted alfalfa. The constant movement of the sample bags in AD1 did not influence their degradability. The degradability values obtained for the modified AD2 were similar to those of AD1. The AD1 instrument was older than the AD2 one, and, as can be noticed from the detected motion reported in Table 1, the movement of the AD1 in its original format was more erratic. For this reason, a modification of the instrument could be useful to obtain more constant movement and more reliable degradability results.

The two Ankom Daisy$^{\text{II}}$ instruments (AD1 and AD2) presented the same problem: the stopping of the movement of jars. Moreover, they had different rotation speeds, especially when they had been filled.

We observed that the main problem with these instruments is connected to friction, which is caused by the weight of the jars when they are filled with liquid and slows down or even stops the rotation mechanism. It was found that the friction was responsible for the malfunctioning of the instruments for two reasons:

- The weight of the filled jars on the drive rollers caused the jars to slip or the drive belts to slip on the drive pulleys.
- The weight of the filled jars on the free rollers caused friction between the free rollers and the supporting pins, which in turn slowed down or even stopped the rotation of the jars.

Even when new belts had been mounted, the combination of the weight of the jars and the heat inside the incubator (39.5 °C) caused malfunctions after a short time.

Measures were introduced, as suggested by Ankom [42], to ensure enough friction between the outside surface of the jars and the rollers. Rough-surface tape was applied to the jars at the point of contact with the rollers to create more friction.

It was used to add a delay because the two instruments had different speeds due to structural differences. When filled with 2 L of water or RF, the jars showed a marked slowdown. Very long delays were measured in the original instruments, with the shortest delay being 46.5 %.

Jars 1 and 2 showed lower speed delay values (%) than jars 3 and 4 in AD1, while just jar 1 slowed down in AD2. Jars 1 and 2 sometimes stopped turning and then just

moved slowly. The delay of jars 3 and 4 was shorter, and they were never observed to have stopped, thus a constant sliding was assumed.

After the modification of AD1, its speed increased. This rph difference between the two versions was due to the installation of a new drive pulley, which had a larger diameter and therefore caused the two drive rollers to turn faster.

The delay (%) in the modified ADs$^{II}$ instrument, with respect to the empty jars, was significantly shorter than in the original version, thus indicating that the jars rotated constantly (Table 3), but the delay between jars in the modified system was still different. Therefore, an improvement in reliability was obtained with the introduced changes, although, due to the high number of measured rph, the short delays were also significantly different in the modified system. The delays were also negative due to the variability of the dimensions of the pulley and rollers, which were not reassembled in exactly the same positions on the two AD$^{II}$s during the various repetitions of the experiments. The modified delays ranged between $-4.5$ and $2.6$ %.

The changes proposed here are suggested with a view to preserving the original instrument as much as possible. The introduction of a toothed wheel and belt to replace the current belts would surely be the best solution, and inexpensive kits are already available on the market.

## 5. Conclusions

The degradability results show that movement of the bags is not always critical to achieve optimal degradability. However, an instrument whose operation was as regular as possible would eliminate the doubts related to inaccurate instrumentation, thereby simplifying the study of any variability obtained in the degradability tests.

The modifications introduced in our experiment were able to stabilize the rotation of the jars, thus making the ADs$^{II}$ more regular. A simple and inexpensive solution has been proposed to solve some reliability problems with the AD$^{II}$ incubator.

Our experimentation has verified that the reliability and accuracy of the AD$^{II}$ incubator have improved to a great extent over time. However, this functional reliability has not necessarily resulted in an improvement in the degradability of feedstuffs compared with less efficient instrumentation.

However, perfect functionality of the instrument is necessary to ensure a scientific and standardized procedure that would allow the reproducibility and comparison of results on feed degradability; modifications of the instrument are thus suggested.

**Author Contributions:** Conceptualization S.T., C.S. and S.B.; Instrument modifications S.B.; Data curation S.B.; Formal analysis S.T., S.G.P., S.M. and C.S.; Writing-original draft preparation, S.T., S.G.P., S.M., C.S. and S.B.; Supervision, S.T., C.S. and S.B.; All authors have read and agreed to the published version of the manuscript.

**Funding:** This research received no external funding.

**Conflicts of Interest:** The authors declare no conflict of interest.

## Appendix A. Modification Introduced to Ensure a Constant Speed of the Jars

*Appendix A.1. Drive Belt, Pulley, and Roller Slipping*

Four drive belts were adopted to solve this problem. Three pieces were manufactured or modified, namely, a new drive pulley with four tracks to fit four drive belts and two drive rollers to fit two drive belts.

a.      A new drive pulley with tracks was constructed to fit four drive belts, using a piece of polycarbonate on which four tracks were cut to accommodate the four drive belts, as shown in Figure A1. Polycarbonate seemed to be the most suitable material because the inner surface of the tracks remained rougher when this material was used. A 1 cm piece of a 4 mm bolt was used as a screw set. A 3.2 mm hole was drilled and filleted at a depth of 4 mm, as shown in Figure A1.

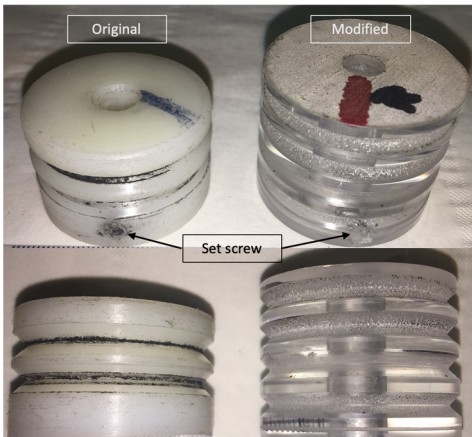

**Figure A1.** The original and modified drive pulleys.

b.   Two drive rollers were obtained to fit the two drive belts by modifying the two original drive rollers in AD$^{II}$. Two other tracks were carved into the original drive roller, as shown in Figure A2, near the original tracks, to position the second drive belt.

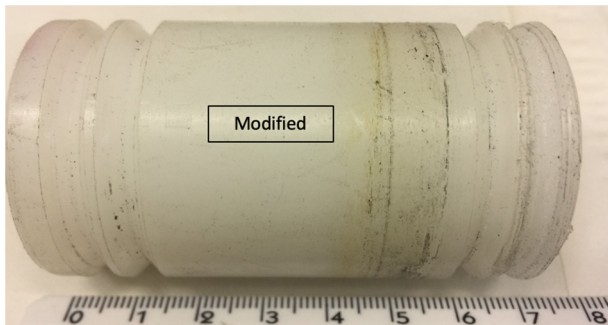

**Figure A2.** Modified drive roller.

It was important to position the double drive belt in such a way as to control the lower jars on the two rearmost guides of the modified drive roller in order to prevent them from rubbing against the bottom of the lower jars (Figure A3).

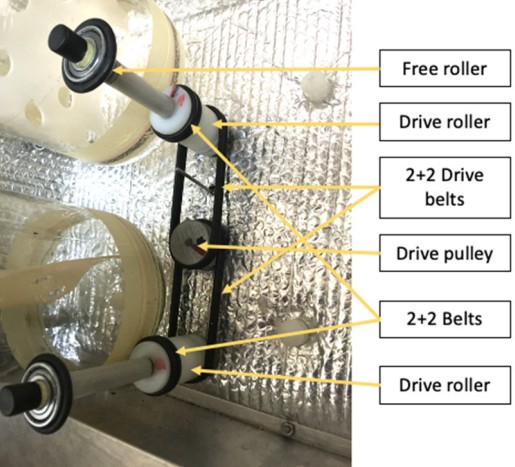

**Figure A3.** Correct positioning of the belts and drive belts.

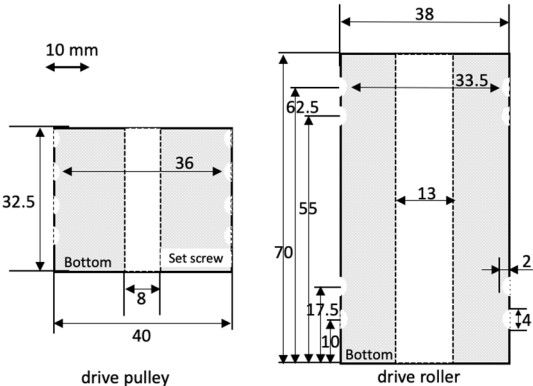

**Figure A4.** Technical drawings with the measurements of the made or modified pieces used to reproduce and adapt the pieces to AD$^{II}$.

*Appendix A.2. Preventing Free Roller Slipping*

The free rollers rotated freely on pins, but the weight of the jars slowed down or even stopped the rotation. This issue was solved by replacing the 10 original free rollers with ball bearings. Ball bearings (ext $\varnothing$ = 35 mm; width = 15 mm; int $\varnothing$ = 13 mm) that are suitable for replacing the original free rollers are available on the market at a low price. The 13 mm internal diameter is larger than the original one, but it is sufficient to introduce a paper cylinder as an adapter to overcome this problem, as shown in Figure A5. Moreover, the original belt was found to fit well with the two central ball bearings, but in order to keep the belt in place during use, a track had to be engraved on the outside of the ball bearing or a strong glue had to be used to hold the belt in place. In this way, it was possible to avoid the use of belts.

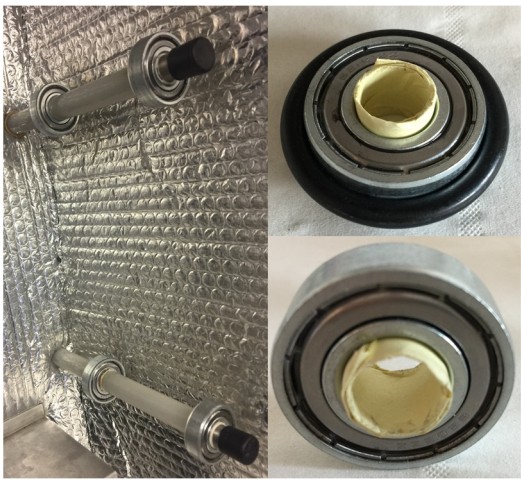

**Figure A5.** Ball bearings used to replace free rollers.

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
