# Peer review of "The Effect of the Stirring Speed on the In Vitro Dry Matter Degradability of Feeds"

_fermentation, doi:10.3390/fermentation9010056_

Round 1

Reviewer 1 Report

1.     The experiment studied the effect of stirring speed on dry matter degradability. However, the title mentioned "feeds degradability".  “feeds” need to be revised to “ feed dry matter".

2.     Lines 91-99: The particle size of the feed samples, the feeding management of the cattle as the rumen fluid donors and the make-up of the buffer solution need to be described in detail.

3.     Line 94: Nylon bags are usually used to determine the ruminal degradability of feeds. It is not suitable that the experiment used Ankom F57 bags to determine the feed degradability because of the much smaller pore size of the Ankom F57 bags than nylon bags.

4.     Lines 107-114: The rotation speeds as treatments need to be clearly described.

5.     Line 123; The calculation of true dry matter degradability seems to be wrong.

6.     The experiment only evaluated the effect of stirring speed on dry matter degradability, but not on the degradability of main nutrients such as crude protein, organic matter, neutral detergent fiber and acid detergent fiber. Therefore, the usefulness of the conclusions of the experiment is limited.

Author Response

English revision has been performed as request.

Reviewer 2 Report

I read with interest your manuscript and have some edits and questions. I found the manuscript very interesting
Abstract
17: ..the apparent and true dry matter degradability (ADMD and TDMD) of grass hay….

Introduction

All in all, the introduction was too long and needs to be revised.

36: that occurs within was largely responsible of fermentation and degradability processes.

63: Indications were given on digestion time. what does this mean?

Materials and Methods

75-76: Please supplement the model and manufacturer of the two ADll

93: dried and milled were weighed(0.5g)  how many milled? 2.5mm?

95: fluid(RF), which collected from ...

94-96: It means jar 1 with rumen fluid, jar 3 with buffer solution?

Table 1

Rotation parameters(N=23690), what are the units? rpm or cm?

Rotation speed (rph) 38.4D  59.4B  47.7C  66.9A

a,b P=0.05, A,B,C,D P=<0.01:based on Tukey’s test on the same row.

Abbreviations: AD1, Ankom Daisyll 1; AD2, Ankom Daisyll 2.

134-135: The equation Delayi (%)=

145: ...was used to evaluate the effects of modification (original vs modified)

153-155: the average values of grass hay, barley meal, pelleted alfalfa, straw, corn silage and TMR were45.2%, 70.2%, 46.2%, 24.7%, 56% and 55.4%, respectively.

177: very high(32.2÷-0.5%), what was mean?

179-180: ...on rotation speed (mean and standard deviation, mean±SD)

Table 2

...48 hours with different content (empty, 2 L of water or rumen fluid(RF), 2 L of RF mixed with buffer solution plus sample bags; )

211: speed was due to the small

220: In table 3 delays were reported among the original and modified ADll incubators. All the comparisons were significantly different

Table 3

Delete the comments”Abbreviations: RF, rumen fluid; AD1, Ankom Daisyll 1; AD2, Ankom Daisyll 2.”

282: The delay of jars 3 and 4 was smaller and during the checks it had ...

A suggestion for Table 1, 2, and 3

It was recommended to remove the vertical line in the middle.

Conclusions

So, what is your recommendation? Should we modified the ADll incubators, based on your data?

References  

According to the guideline for the journal, modify the format of the references .

Author Response

You can find all the details in the file in attach.

Round 2

Reviewer 1 Report

I have no further comments.